# TiO_2_ (Core)/Crumpled Graphene Oxide (Shell) Nanocomposites Show Enhanced Photodegradation of Carbamazepine

**DOI:** 10.3390/nano11082087

**Published:** 2021-08-17

**Authors:** Han Fu, Kimberly A. Gray

**Affiliations:** Department of Civil and Environmental Engineering, Northwestern University, 2145 Sheridan Road, Evanston, IL 60208, USA; hanfu2023@u.northwestern.edu

**Keywords:** environmental photocatalysis, PPCPs removal, carbamazepine photodegradation, spray drying technology, core-shell structured catalyst, titanium dioxide graphene composite

## Abstract

The presence of pharmaceuticals and personal care products (PPCPs) in aquatic systems is a serious threat to human and ecological health. The photocatalytic degradation of PPCPs via titanium oxide (TiO_2_) is a well-researched potential solution, but its efficacy is limited by a variety of environmental conditions, such as the presence of natural organic macromolecules (NOM). In this study, we investigate the synthesis and performance of a novel photoreactive composite: a three-dimensional (3D) core (TiO_2_)-shell (crumpled graphene oxide) composite (TiGC) used as a powerful tool for PPCP removal and degradation in complex aqueous environments. TiGC exhibited a high adsorption capacity (maximum capacity 11.2 mg/g, 100 times larger than bare TiO_2_) and a 30% enhancement of photodegradation (compared to bare TiO_2_) in experiments with a persistent PPCP model, carbamazepine (CBZ). Furthermore, the TiGC performance was tested under various conditions of NOM concentration, light intensity, CBZ initial concentration, and multiple cycles of CBZ addition, in order to illustrate that TiGC performance is stable over a range of field conditions (including NOM). The enhanced and stable performance of TiCG to adsorb and degrade CBZ in water extends from its core-shell composite nanostructure: the crumpled graphene oxide shell provides an adsorptive surface that favors CBZ sorption over NOM, and optical and electronic interactions between TiO_2_ and graphene oxide result in higher hydroxyl radical (•OH) yields than bare TiO_2_.

## 1. Introduction

A variety of pharmaceuticals and personal care products (PPCPs) are released into waterways daily from wastewater effluent discharges associated with homes, municipalities, and industries [1,2,3]. Conventional wastewater treatment technologies, which are primarily based on biological processes, may only reduce some PPCPs but do not remove all of them completely [4,5]. As a result, many PPCPs are detected at trace levels (ng to µg/L) in municipal wastewater effluents, as well as in surface waters [6,7]. Their wide distribution in natural waters, possible bioaccumulation in food webs, and endocrine-disrupting effects threaten both ecological and human health [8,9,10,11]. Carbamazepine (CBZ), a widely used pharmaceutical to treat epilepsy and depressive disorders, is one example of persistent PPCPs. Several studies have demonstrated that CBZ is difficult to biodegrade and is routinely detected in water systems [12,13,14]. Thus, CBZ is a good model of PPCPs removal under environmental conditions. 

There are numerous reports in the literature that photocatalytic oxidation processes mineralize CBZ and many other PPCPs [15,16,17]. TiO_2_ is a widely used heterogeneous and wide bandgap photocatalyst that shows a robust performance and has a low cost. Various studies report the photocatalytic degradation of CBZ by reactive oxygen species (ROS) generated by TiO_2_ under UV radiation [18,19,20]. However, the influence of environmental factors, such as the presence of natural organic macromolecules (NOM), is often overlooked in evaluating the performance of novel nanomaterials for contaminant removal applications in water. NOM is a complex mixture of organic materials derived from a range of biopolymers, such as tannins, lignins, polysaccharides, proteins, and lipids, and is a common component in surface and ground waters present at concentrations that are orders of magnitude greater than CBZ or other PPCPs. NOM may interfere with contaminant adsorption and photodegradation through competitive interactions at reactive surfaces, by scavenging ROS and attenuating light radiation [21,22]. Hence, it is necessary to develop a TiO_2_-based nanomaterial that can degrade CBZ and other PPCPs efficiently and stably under realistic environmental conditions.

Graphene, an allotrope of carbon consisting of a single-atom-thick hexagonal lattice of sp^2^-bonded carbon atoms, has attracted tremendous research attention. The physicochemical properties make graphene and its composites attractive for PPCP removal and other environmental applications [23,24,25]. Some studies demonstrate that a composite of TiO_2_ and graphene/reduced graphene oxide show an improved photodegradative capacity to eliminate CBZ compared to TiO_2_ alone [26,27,28]. In such composites, graphene may reduce the electron–hole recombination, increase ROS generation, and extend the photo response of TiO_2_, thereby enhancing the photocatalytic activity of the nanomaterial to degrade pollutants. 

In addition, graphene-based nanomaterial surfaces show a strong affinity for organic compounds and therefore can be employed as an environmental adsorbent to remove PPCPs in water [29,30,31]. Unfortunately, due to the strong van der Waals interaction between the two-dimensional (2D) sheets, the irreversible stacking of graphene and graphene oxide occurs in water which reduces the exposed surface area and limits the adsorption efficacy [32,33]. A promising strategy to resolve this aggregation phenomenon is to assemble three-dimensional (3D) adsorbents from the graphene/graphene oxide 2D nanosheets [34]. 

Recently, we fabricated 3D crumpled graphene balls (CGBs) that resist the stacking of graphene nanosheets and display an enhanced adsorption performance for CBZ and other PPCPs removal compared to GO and granular activated carbon [35]. More importantly, the robust adsorption performance of CGBs is not adversely affected by NOM and other environmental conditions (e.g., pH, ionic strength, water hardness, and alkalinity). Herein, we report a novel approach to combine CGBs with TiO_2_ in order to create an adsorptive and photoactive composite designed for PPCP removal.

In the present study, we synthesized a unique core (TiO_2_)-shell (graphene oxide) composite (TiGC) via spray drying, which exhibits a superior photodegradation of CBZ in comparison to TiO_2_. We characterized the TiGC structure and investigated the CBZ adsorption and photodegradation reaction at various TiO_2_ and graphene oxide ratios. Upon identifying an optimum TiO_2_/GO ratio, we further detailed the TiGC efficiency under various field conditions, such as light intensities, CBZ initial concentrations, the presence of NOM, and multiple cycles of CBZ adsorption/degradation. Finally, we also quantified ROS yields in order to explain the enhanced performance of TiGC. 

## 2. Methods and Materials

### 2.1. Materials and Reagents

Carbamazepine was purchased from Sigma-Aldrich (St. Louis, MO, USA). Titanium dioxide AEROXIDE P25 (P25) was provided by Evonik Industries (Essen, Germany). Graphene oxide (GO) was synthesized through a modified Hummer’s method as previously reported (detailed synthesis method can be found in Appendix A) [36,37]. Formic acid (EMD^TM^ 98%, Burlington, MA, USA) and ammonia hydroxide (Honeywell 5.0 M solution, Charlotte, NC, USA) were used to adjust pH. Suwannee River fulvic acid (SRFA, code 2S101F), used to model the effect of NOM, was acquired from the International Humic Substance Society (Denver, CO, USA). The elemental composition and functional group analysis are found on IHSS website [38]. 

### 2.2. Synthesis and Characterization

TiGC was made from the GO and P25 suspension by spray drying method using a spraying instrument (Buchi Nano-spray Dryer B-90, BUCHI Lab Equipment AG, Flawil, Switzerland), as described in our previous study [35]. A selected amount of P25 particle (600 mg, 300 mg, 120 mg, or 60 mg corresponding to a TiO_2_:GO weight ratio 10:1, 5:1, 2:1, and 1:1) was first prepared in 140 mL purified water and sonicated for 30 min; 60 mL GO stock solution (1 mg/mL) was then added and continuously sonicated for another 30 min in order to obtain a well suspended GO and P25 mixture. It was determined that a final concentration of 0.3 mg/mL GO in the 200 mL final suspension avoids GO stacking during the nano-spray synthesis. The pH of the mixture was adjusted to 6.5 and sprayed at 95 °C via the Buchi Nano-spray Dryer B-90 (nozzle size 4 μm, part #*051747*) while continuously stirring the suspension. The particles were collected in the particle collection chamber at the bottom of the instrument. The collected particles were then annealed at 150 °C for 60 min to remove moisture and impurities on the surface. 

TiGC was characterized by a variety of techniques. SEM images were acquired with a Hitachi SU8030 (Hitachi High-Technologies Corporation, Tokyo, Japan) and EDS analysis was conducted via AZtec X-max 80 SDD EDS detector (Oxford Instruments PLC, Oxfordshire, UK). TEM images were gathered with the JEOL JEM-2100F Field Emission Electron Microscope (JEOL Ltd., Tokyo, Japan). Nitrogen adsorption and desorption analyses were conducted with a Micromeritics 3Flex surface Area Analyzer and its related software MicroActive™ (Micromeritics Instrument Corporation, Norcross, GA, USA). X-ray photoelectron Spectroscopy (XPS) was performed on the Thermo Scientific ESCALAB 250Xi (Thermo Fisher Scientific Inc., Waltham, MA, USA). FTIR spectroscopy measurements were made with a Nexus 870 spectrometer (absorbance mode, 4 cm^−1^ resolution, 4000–400 cm^−1^ wavelength range, 64 scans, Thermo Fisher Scientific Inc., Waltham, MA, USA). The UV-vis spectrum of TiGC and TiO_2_ were measured via an Eppendorf BioSpectrometer® basic (Eppendorf AG, Hamburg, Germany).

### 2.3. Photodegradation Experiment 

Experiments on the photodegradation of carbamazepine were performed through batch experiments in Milli-Q water (Milli-Q RG QPAK 1 column, water *resistivity* 18.2 MΩ-cm at 25 °C, Merck Millipore Inc., Burlington, MA, USA). A stock solution of 1 g/L carbamazepine is first prepared and then stored at 4 °C for further dilution. Similarly, NOM solutions at 10 mg/L and 20 mg/L as SRFA were prepared and stored at 4 °C. Stock solutions were stirred for 30 min at room temperature before usage. Illumination was supplied by UVP® 100-Watt Ultraviolet Mercury Lamp (model #*B-100A*, Analytik Jena AG, Jena, Germany), and Newport 1000 W Ozone Free Xenon Arc Lamp (model #*6271 Ozone Free*) with controllable power output 500 and 1000 W (Newport Corporation, Irvine, CA, USA). The light wavelength intensity was monitored by a SpectriLight ILT950 spectroradiometer (International Light Technologies, Peabody, MA, USA). A total of 50 mL CBZ solution at different initial concentrations (1 mg/L to 3 mg/L) was prepared in a 100 mL beaker and the pH of the solution was adjusted to 7.5 ± 0.1 via formic acid or ammonia hydroxide (Honeywell 5.0 M solution). Then, 0.2 mg/mL of catalyst (TiGC or TiO_2_) was added, and the suspension was stirred in dark for 30 min to reach the adsorption equilibrium. After the light was switched on, the suspension was stirred at 300 rpm. At periodic intervals (every 30 min), the suspension was sampled via a syringe (BD 1 mL TB syringe) and 0.2 µm PTFE syringe filter (Whatman^TM^, Whatman PLC, Buckinghamshire, UK). The degradation of CBZ reaction by-products was evaluated via UV-vis spectrum (Eppendorf BioSpectrometer^®^ basic, Eppendorf AG, Hamburg, Germany) at wavelength 256 nm (aromatic structure) and 280 nm (amide group).

CBZ concentration was quantified by HPLC-MS/MS (QExactive, Thermo-Fisher Scientific Inc., Waltham, MA, USA) and a Thermo-scientific BDS Hypersil C18 column with the following protocol: 10 µL of the sample was injected by pumping the mobile phase of deionized water and acetonitrile (both mobile phases contain 0.1% formic acid) at a speed 0.4 mL/min. The MS was operated with electrospray ionization in positive and negative polarity modes and analyte levels were determined from calibration standards based on linear regression calculation. Using positive ionization mode, the exact mass of CBZ is 237.1, and its retention time is 6.5 min.

We chose two different models to evaluate CBZ photodegradation kinetics: a pseudo first order reaction model and the Langmuir-Hinshelwood (L-H) model. The linearized form of pseudo first order reaction is: (1)$$\ln{\left[CBZ\right]}_{0}-\ln{\left[CBZ\right]}_{t}=k_{1}t$$
where [*CBZ*]_0_ and [*CBZ*]*_t_* are the initial CBZ concentration and concentration at time *t* (µg L^−1^), respectively. *k*_1_ is the pseudo first-order reaction constant (min^−1^). The linearized form of L-H model is:(2)$$\frac{1}{r_{0}}=\frac{1}{k_{L-H}K_{L-H}}\times \frac{1}{{\left[CBZ\right]}_{0}}+\frac{1}{k_{L-H}}$$
where *r*_0_ is the initial photocatalytic degradation rate (determined via initial rate method [39,40,41]), *k_L-H_* is the *L*-*H* reaction constant (µg L^−1^ min^−^^1^), *K_L-H_* is the *L*-*H* adsorption constant (L µg ^−1^), and [*CBZ*]_0_ is the initial CBZ concentration (µg L^−1^).

### 2.4. Adsorption Experiment 

Adsorption isotherms were performed through batch adsorption experiments in Milli-Q water or NOM solutions, as described in our previous paper [35]. The initial CBZ concentrations varied from 1 to 5 mg/L and the pH of the prepared solutions was adjusted to 7.5 ± 0.1 with formic acid or ammonia hydroxide. The prepared solutions were added to 20 mL Teflon-lined screw-top glass vials and the TiGC (adsorbent dosage 0.1 mg/mL) was then added. All sample vials were mixed on a VWR incubator orbital shaker (200 rpm, 25 °C, VWR International, Radnor, PA, USA) for 24 h to reach equilibrium. After 24 h of shaking, the vials were sampled via a syringe (BD 1 mL TB Syringe, Franklin Lakes, NJ, USA) and 0.2 µm PTFE syringe filter (Whatman^TM^, Whatman PLC, Buckinghamshire, UK). The CBZ concentrations in solution were then quantified with the same methods as in the photodegradation experiment. The Langmuir model was utilized to describe the adsorption behavior of TiGC:(3)$$q_{e}=\frac{q_{m}K_{L}c_{e}}{1+K_{L}c_{e}}$$
where *q_e_* (mg/g) is the adsorbed MP per adsorbent mass at equilibrium, *C_e_* (mg/L) is the MP solution concentration at equilibrium, and *q_m_* (mg/g) and the *K_L_* (L/mg) are the maximum adsorption capacity and adsorption affinity parameter, respectively.

### 2.5. CBZ Recovery after Adsorption/Photodegradation Experiment 

In order to determine the amount of adsorbed CBZ remaining on the TiGC/CGB at the end of photodegradation experiments, a solvent wash method using acetonitrile (ACN) was employed to recover adsorbed CBZ [42]. Used TiGC/CGB was first collected via vacuum filtration (0.22 μm Fisherbrand™, *model#SA1J789H5*, Thermo Fisher Scientific Inc., Waltham, MA, USA). Filtered TiGC/CGB was dried at 60 °C overnight. The dried TiGC was suspended in ACN (Sigma-Aldrich, St. Louis, MO, USA, 99.9%) with an approximate concentration 0.2 mg/mL and stirred on a VWR incubator orbital shaker (200 rpm, 25 °C, VWR International, Radnor, PA, USA) for 24 h to complete the CBZ recovery. The TiGC/CGB was then separated via centrifuge (Effendorf Model #5810, Eppendorf International, Hamburg, Germany) at 9000 rpm for 15 min and the recovered CBZ in the liquid phase was analyzed via HPLC-MS/MS.

### 2.6. Reactive Oxygen Species (ROS) Measurement

ROS was quantified using the appropriate absorbance and fluorescence molecular probes. All tests were performed in 96-well microtiter plates, and molecular probes were detected using the Gemini EM fluorescence microplate reader (model # *BZBLKU765*, Biotek, Winooski, VT, USA). Illumination was provided by 1000 W Xenon lamp. Three replicates were performed for each measurement. The control experiments were performed in the absence of catalyst under the same irradiation conditions. First, hydroxyl radical (•OH) production was assessed after reaction with coumarin-3-carboxylic acid (3-CCA, 98%, Alfa Aesar, Haverhill, MA, USA) to form fluorescein [43]. Ten µL of 100 µM 3-CCA, 10 µL of catalyst mixture, and 80 µL of purified water were added to each well in a microtiter plate. After 30 min of irradiation, the pH of the mixture was adjusted to 9.5. The reaction of 3-CCA and hydroxyl radical produced 7-hydroxycoumarin-3-carboxylic acid (7-HO-3-CCA). The fluorescence due to the excitation of 7-HO-3-CCA was measured (ex/em = 387 nm/447 nm). Second, superoxide anion (O_2_•^−^) was determined using 2,3-bis (2-methoxy-4-nitro-5-sulfophenyl)-2H-tetrazolium-5-carboxanilide sodium salt (XTT sodium salt, Sigma-Aldrich, St. Louis, MO, USA) [44]. Twenty µL of 1 mM XTT, 20 µL of nanoparticle mixtures, and 160 µL of purified water were added to plate wells and then exposed to 1000 W Xenon lamp irradiation for 30 min. The reaction of XTT with superoxide anion produced XTT formazan, which was detected by measuring its absorbance at 470 nm. Third, hydrogen peroxide (H_2_O_2_) production was detected using phenol red [45]. Ten µL of the catalyst mixture was added to 90 µL of LMW in a well plate and irradiated for 30 min. Ten µL of 0.77 M NaOH, 10 µL of 1 g/L phenol red (Sigma-Aldrich, St. Louis, MO, USA), and 10 µL of 0.5 mg/mL horseradish peroxidase (Type II, salt-free powder, Sigma-Aldrich St. Louis, MO, USA) were then added to each well and mixed thoroughly. The absorbance at 610 nm was measured to detect the oxidation product of phenol red and hydrogen peroxide. The absorbance for the control is due to the red color of the phenol red dye, and changes in absorbance at 610 nm are attributed to the purple color of the reaction product.

## 3. Results and Discussion

### 3.1. TiGC Characterization

Figure 1 shows the morphology of TiGC at a TiO_2_:GO ratio of 2:1 (additional SEM images for other TiO_2_:GO weight ratios are shown in Appendix A). The SEM images in Figure 1a,b show the crumpled-paper-ball-like structure with a particle size range of 0.5–2 µm. The surface is covered by wrinkled graphene oxide sheets and no TiO_2_ nanoparticles were observed at the surface. When we increase the TiO_2_ amount in the composite, more TiO_2_ particles are exposed at the surface and the TiGC particle size increases substantially (Appendix A) due to more TiO_2_ addition. TEM images (Figure 1c,d) further confirm the core-shell structure of TiGC, in which a shell formed by the folds and wrinkles of the GO sheet encapsulates clusters of nano-TiO_2_. The HRTEM image (Figure 1e) provides an enlargement of the outlined area of Figure 1d, revealing the lattice fringes (with 0.35 nm interplanar spacing) of the (101) plane of the TiO_2_ anatase phase. The EDS analysis (Figure 1f) illustrates the compositional analysis of TiGC with carbon, oxygen, and titanium as the major components. A small portion of sulfur (0.6%) is also observed, which is a common impurity of GO due to the utilization of sulfuric acid during GO synthesis. A spatial plot of the elemental analysis along the two line scans (shown in Figure 1f) reveals that titanium is concentrated in the core of the structure. Furthermore, we studied the morphology of TiGC (TiO_2_:GO weight ratio 2:1) after the photodegradation experiments (SEM images shown in Appendix A, TEM images shown in Appendix A) and found that the majority of TiGC retains its 3D core-shell structure, although in a few cases (circled in Appendix A), we observed that some TiO_2_ nanoparticles may migrate from the core to decorate the TiGC surface.

XPS and FTIR were employed to investigate the surface chemical characteristics of TiGC (Figure 2a–d). The wide scan XPS spectrum (Figure 2a) reveals that oxygen, carbon, and titanium are major components of TiGC. The C 1s XPS spectrum (Figure 2b) further demonstrates the presence of the oxygen-containing functional groups, such as C–O–C and C=O. Two titanium peaks (Figure 2c) indicate the presence of oxidized titanium. The FTIR spectrum (Figure 2d) confirms the same observations as the XPS spectrum (Figure 2b): –OH, C=O, C–O, and C–O–C peaks are observed. Overall, TiGC retains abundant oxygen-containing functional groups after synthesis. The nitrogen adsorption–desorption isotherms of both TiO_2_ and TiGC exhibited the IUPAC Type IV isotherm curve (Figure 2e) that is consistent with a micro/mesoporous material. The specific surface area of TiGC is 107.92 m^2^/g, which is more than 50% larger than TiO_2_ (65.38 m^2^/g). Hysteresis is observed in the TiGC adsorption–desorption curve, indicating the presence of mesopores and micropores. The pore size distribution (Figure 2f) further confirms the existence of both micropores (<2 nm) and mesopores (2–100 nm) for TiGC. Compared to TiGC, the aggregates of TiO_2_ have mesopores (2–100 nm) and macropores (>100 nm) as the dominant pore size. 

### 3.2. Effect of Varying TiO_2_ and GO Ratios in TiGC on CBZ Photodegradation 

The CBZ degradation by TiO_2_, crumpled graphene ball (CGB, with no TiO_2_ addition during synthesis), and TiGC (varying TiO_2_ and GO weight ratios from 10:1 to 1:1) under 1000 W Xenon light illumination are compared in Figure 3a. The experiments were conducted in two stages: 30 min under dark conditions (to measure adsorption) followed by a 2 h illumination (to investigate the photodegradation performance). The control experiment shows that the photolysis of CBZ is negligible. TiO_2_ shows negligible adsorption in the dark, but is capable of degrading CBZ under light illumination (90% removal within 2 h). CGB shows >90% CBZ removal via adsorption in the dark, but no further photodegradation of adsorbed and solution-phase CBZ is observed under light illumination (adsorbed CBZ is confirmed by the recovery test in Figure 3b). Rapid adsorption kinetics are also consistent with our previous research, which utilized CGB as the adsorbent for various PPCPs removals [35].

Compared to CGB and TiO_2_ alone, the composite TiGC exhibits the combined features of adsorption and photodegradation. As illustrated in Figure 3a, the relative adoption or photodegradation capacity is tuned by the TiO_2_:GO ratio in the composite. In order to determine the optimal ratio, we first performed adsorption isotherm tests (detailed adsorption curves and calculated *q_m_* and *K_L_* are shown in Appendix A). Figure 3c summarizes the Langmuir maximum adsorption capacity, *q_m_*, to show that increasing the GO proportion in the composite increases the adsorption capacity of the composite; the largest *q_m_* value is at a TiO_2_:GO ratio of 1:1. Then, we compare the CBZ photodegradation by TiO_2_ and TiGC using the reaction constants of two models: *k*_1_ (pseudo first-order reaction model, Figure 3d) and *k_L-H_* (Langmuir–Hinshelwood L-H model, Figure 3e). Details are provided in Appendix A. The results for *k*_1_ (Figure 3d) indicate an enhanced CBZ photodegradation of TiGC at three TiO_2_/GO ratios: 10:1, 5:1, and 2:1. Moreover, the results of *k_L-H_* (Figure 3e) display a superior performance of all TiGC materials compared to bare TiO_2_, and among them, the TiO_2_:GO ratios of 2:1 and 1:1 exhibited the best performance.

In both models, TiGC (at all TiO_2_:GO ratios, except 1:1 in the pseudo first-order reaction model) showed a better performance than bare TiO_2_, which is consistent with other reported TiO_2_–graphene composites [26,27,28,46,47,48,49,50]. The enhancement can be attributed to three major reasons: first, the ROS generation yield is enhanced by interactions between TiO_2_ and GO (confirmed in Section 3.7); second, adsorptive sites created by the GO shell shorten the diffusion time of ROS to the target CBZ; third, the relatively low performance of TiO_2_ may be caused by both the aggregation of nano-sized TiO_2_ in water and the absorbance of a narrower range of ultra-band gap light [27,28]. The L-H model, however, assumes surface reactions between sorbed CBZ molecules and a nearby bound species; in this case, a bound ROS [40]. The CBZ degradation mechanism on TiGC, however, may be more complex. For instance, the mechanisms and yields of ROS generation may differ between TiO_2_ and TiGC, as discussed below. Overall, considering the *q_m_*, *k*_1_, and *k_L-H_*, we selected 2:1 as the optimal TiO_2_:GO ratio in TiGC synthesis and conducted further investigation.

We further investigated the fate of adsorbed CBZ in order to determine whether adsorptive sites can be regenerated via photodegradation for CGB and TiGC at two TiO_2_:GO ratios: 2:1 and 1:1. Figure 3b categorizes CBZ (initial concentration [*CBZ*]_0_) into three groups at different times in the photodegradation experiment: (i) CBZ remaining in the aqueous phase, [*CBZ*]_aq_ ([*CBZ*]_aq_ = [*CBZ*]_0_ at *t* = 0); (ii) CBZ adsorbed, [*CBZ*]_ad_ (recovered via the solvent wash method described in Section 2.5); (iii) CBZ photodegraded, [*CBZ*]_photo_. In the case of CGB, the adsorption of CBZ occurs in the first 30 min and there is no further change in [*CBZ*]_photo_ and [*CBZ*]_ad_ for the 2 h of light illumination. In the case of 2:1 TiO_2_:GO ratio, after 30 min of adsorption in the dark, [*CBZ*]_ad_ reaches approximately 23% of [*CBZ*]_0_; after 2 h of light illumination, [*CBZ*]_ad_ is 10% of [*CBZ*]_0_ (meaning 50% of adsorbed CBZ in the first 30 min is photodegraded), while the total [*CBZ*]_photo_ is 84% of [*CBZ*]_0_, and [*CBZ*]_aq_ is 6% of [*CBZ*]_0_. In the case of the TiO_2_:GO ratio of 1:1, [*CBZ*]_ad_ is >80% of [*CBZ*]_0_ at 30 min. However, at the end of the 2 h of illumination, 70% of [*CBZ*]_0_ still remains adsorbed (meaning only 12.5% of adsorbed CBZ in the 30 min adsorption is photodegraded), 25% of [*CBZ*]_0_ is photodegraded, and 5% of [*CBZ*]_0_ remains in the aqueous phase. These results indicate that the regeneration of the adsorptive sites (surface reactions) and the photodegradation of CBZ in water may occur simultaneously, but with different kinetics. The reaction kinetics between ROS and CBZ are affected by both their relative rates of diffusion and the ROS formation mechanism [28,46,47]. The different reaction kinetics of free CBZ and bound CBZ may also help to explain the case of the TiO_2_:GO ratio of 1:1, where it takes a long time (a lower pseudo first model reaction constant) to degrade CBZ, since there are much more bound CBZ than free CBZ. Finally, these results indicate that there is an optimum TiO_2_:GO ratio that is defined by the relative rates of adsorption and photodegradation.

Finally, we evaluated the degree of CBZ mineralization in water (the conversion into inorganic compounds) via a UV spectrum measurement at two wavelengths: 265 nm (representing aromatic rings) and 280 nm (representing amide groups). The results (Appendix A) indicate the reduced absorbance intensities at both wavelengths after the 2 h light illumination, which reflects the destruction of aromatic carbon in the solution and in the case of 2:1 TiGC, suggesting the mineralization of CBZ in the solution phase.

### 3.3. Multiple Cycles of Carbamazepine Additions 

The catalytic efficiency and stability of TiGC (ratio 2:1) was also investigated by repeating the addition of the new CBZ solution for four 2-h reaction cycles (Figure 4). TiGC showed a complete CBZ removal in each cycle, no decrease in CBZ decay rates, and no loss in catalytic activity over the four cycles. In contrast, the rates and extent of the CBZ reaction on bare TiO_2_ declined with each cycle of CBZ addition, suggesting a loss of catalytic activity. Therefore, compared to TiO_2_, TiGC exhibits robust CBZ adsorption and photocatalytic degradation over multiple reaction cycles.

### 3.4. Effect of Illumination Conditions

Since the activity of a photocatalyst is directly affected by the spectrum and intensity of the light source, and since there are various mercury and xenon lamps available in the market, we compared the performance of TiO_2_ and TiGC using three light sources: 100 W UV mercury lamp, 500 W Xenon lamp, and 1000 W Xenon lamp (Figure 5); the spectra of the three light sources are shown in Appendix A. Both TiO_2_ and TiGC show an enhanced photodegradation performance with an increasing light intensity. TiGC shows a superior performance relative to TiO_2_ under the two higher power outputs of the xenon lamp. Due to the core-shell structure, a higher light intensity is necessary to activate the encapsulated TiO_2_ in the core.

The spectra of each lamp (Appendix A) and the light intensity table (insert, Figure 5) show that the 100 W UV lamp primarily supplies UV-A light, whereas the xenon lamps provide a wider wavelength range from UV-A to visible light. The enhanced performance of TiGC under the xenon light suggests that TiGC may be activated by visible light wavelengths due to interactions with both its GO shell, and the UV-vis absorbance spectra of TiGC and TiO_2_ (Appendix A) is consistent with this proposal. Previous studies have reported an enhanced photoactivity of TiO_2_-GO composites under visible light [48,49,50]. The interaction between GO and TiO_2_ extends the photo-response in the visible light range, as shown in Appendix A, by possibly allowing charge injection into the TiO_2_ conduction band, and also by creating band gap states on TiO_2_ that effectively narrow its band gap, leading to visible light activation [48]. In addition, the crumpled structure of the GO shell may promote light scattering into the interior (core) of TiGC, rather than the surface absorbance/reflection of light observed with the layered GO [51].

### 3.5. Effect of CBZ Initial Concentration 

CBZ photodegradation under varying CBZ initial concentrations was investigated and the pseudo first-order reaction constants are summarized in Figure 6. TiGC kinetics are relatively constant over the range of CBZ concentrations (with 10–15% deviation in the reaction constant), whereas the decomposition rate constant of TiO_2_ decreases with an increasing CBZ concentration (more than 80% reduction in the reaction constant). A similar observation was reported by others: a higher CBZ initial concentration suppressed the TiO_2_ rate of degradation [18,52]. Im et al. proposed that, in a TiO_2_ colloidal suspension, excess CBZ molecules accumulate on the TiO_2_ surface, suppressing the light absorbance and ROS generation by TiO_2_ [18]. These phenomena may also account, in part, for why the TiO_2_ performance gradually declined over multiple CBZ additions, as shown in Figure 4. In the case of TiGC, we propose that the GO shell buffers the CBZ reaction. At the same time, since the mechanisms of the CBZ adsorption, reaction, and ROS generation are different for bare TiO_2_ and TiGC, the rate-limiting steps may also differ. 

### 3.6. Effects of pH and NOM 

To further explore the effects of environmental conditions, we first investigated the effect of pH at 6.5 and 8.5 (a typical pH range for surface water), as shown in Appendix A. The results show a negligible change in TiGC performance within the selected pH range. We then tested the effect of NOM on CBZ photodegradation (Figure 7). In natural water systems, NOM is present typically at concentrations that are orders of magnitude higher than CBZ, and exerts competitive effects on both CBZ adsorption and photooxidation. Similar to our previous study [35], we utilized a standardized NOM model, SRFA, which is extracted from the Suwannee River at two concentrations (10 mg/L and 20 mg/L) in order to model the effect of both an average and high TOC level for surface waters. Figure 7a compares the pseudo first-order reaction constants of TiGC and TiO_2_ at the two SRFA concentrations. Increasing NOM concentrations slightly inhibited the TiGC performance; the pseudo first-order reaction constant was reduced by 14% at both SRFA concentrations. In contrast, the inhibitory effect of NOM on TiO_2_ is more pronounced (a 31.8% reduction at 10 mg/L SRFA).

The suppressing effect of NOM on photooxidation operates in a variety of ways: (1) an excess amount of NOM may accumulate at the TiO_2_ surface so as to reduce light absorbance [4,53]; (2) ROS is not selective, and NOM competes with CBZ for ROS [25]. We propose that the GO shell of TiGC buffers NOM effects in two ways: (1) the GO shell works as a “TiO_2_ protector” and hinders the accumulation of NOM molecules at the TiO_2_ surface in order to prevent NOM interference on light absorbance and ROS scavenging; (2) the GO shell serves as the “CBZ capturer”, facilitating CBZ removal and eventual photodegradation, and as demonstrated in our previous work, CBZ adsorption does not diminish over a wide range of NOM concentrations [35]. We confirm this behavior in the case of TiGC, and measured CBZ adsorption at two NOM concentrations in order to find that they are identical to the curve in the case of purified water (no presence of NOM), as shown in Figure 7b (the calculated maximum capacity, *q_m_*, and affinity parameter, *K_L_*, are listed in Appendix A). The values of *q_m_* and *K_L_* do not change much in the presence of NOM, indicating that there is greater selectivity for CBZ than NOM. 

### 3.7. Comparison of ROS Yields between TiGC and TiO_2_


In order to interrogate the interaction between GO and TiO_2_ and its effect on CBZ photodegradation, we measured the ROS generation at 10 mg/L TiO_2_ and TiGC using the absorbance or fluorescence signals of three probe molecules (Figure 8). At the selected time (30 min), TiO_2_ showed more O_2_•^−^ and H_2_O_2_ generation than TiGC, and whereas TiGC yields a 6–7 folds increase in the amount of •OH. Upon the absorbance of light, electrons and holes are created and migrate to the TiO_2_ surface where interactions with oxygen or water generate various types of ROS. The GO shell may help to separate the charges, hinder the recombination of electrons/holes, and enhance the ROS generation [27,28]. In addition, there is interconversion among ROS, and we propose the TiO_2_/GO composite facilitates these interactions leading to greater net •OH yields. Among all three studied ROS, many studies have shown that •OH is the major ROS contributing CBZ degradation [20,21,22,54]. Therefore, the enhanced •OH generation explains the accelerated CBZ photodegradation by TiGC.

## 4. Conclusions

In this study, we synthesize a novel core-shell catalyst, TiGC, and investigate its adsorption and photodegradation ability using CBZ as a model pollutant. Material characterization indicates that a crumpled graphene oxide shell surrounds a TiO_2_ core and, in this study, an optimum ratio of 2:1 TiO_2_:GO was identified. TiGC exhibits an enhanced photocatalytic degradation ability compared to TiO_2_ alone under a variety of conditions, including the presence of NOM, a range of initial CBZ concentrations, and multiple cycles of CBZ addition. TiGC’s photo-reactivity is greater over a broader wavelength of the light. The crumpled GO shell encapsulating the TiO_2_ core enhances the adsorptive and photodegradative performance of the TiGC material in complex ways. The GO adsorbs CBZ selectively in the presence of NOM, concentrating it at its surface. Optical and electronic interactions between GO and TiO_2_ extend the photo-response of TiGC into the visible range and result in much higher •OH yields than TiO_2_ alone, likely due to charge separation, hindered recombination, and facilitated interconversion among ROS. This research lays the groundwork for the development of a self-regenerating adsorptive material for contaminant removal in environmental applications. 

## Figures and Tables

**Figure 1 nanomaterials-11-02087-f001:**
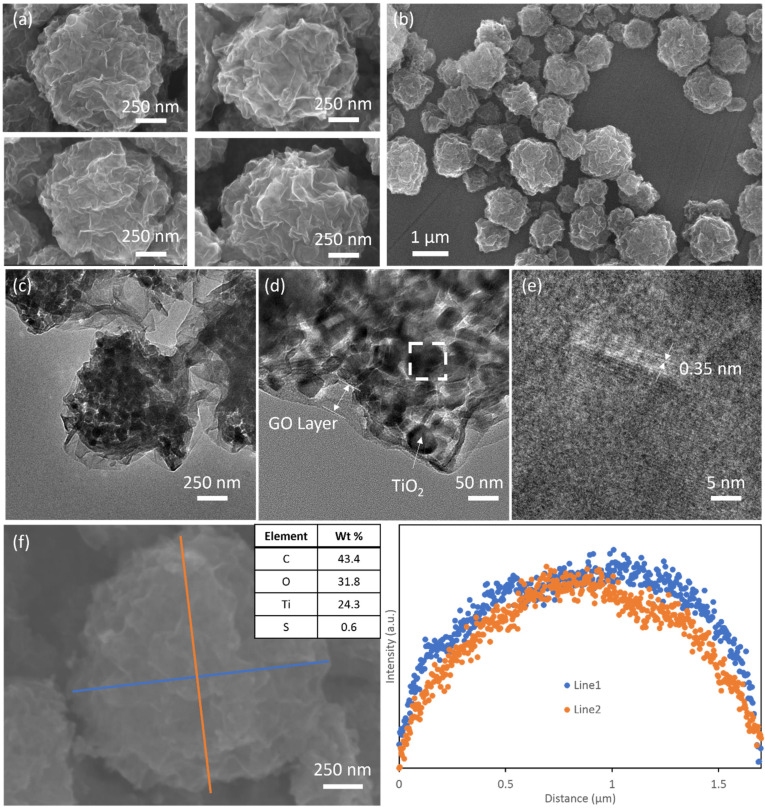
(**a**,**b**) SEM images of TiGC; (**c**,**d**) TEM images of TiGC; (**e**) HRTEM images of TiGC (enlarged area outlined in (**d**)); (**f**) EDS overall element analysis and two line scans of titanium in TiGC.

**Figure 2 nanomaterials-11-02087-f002:**
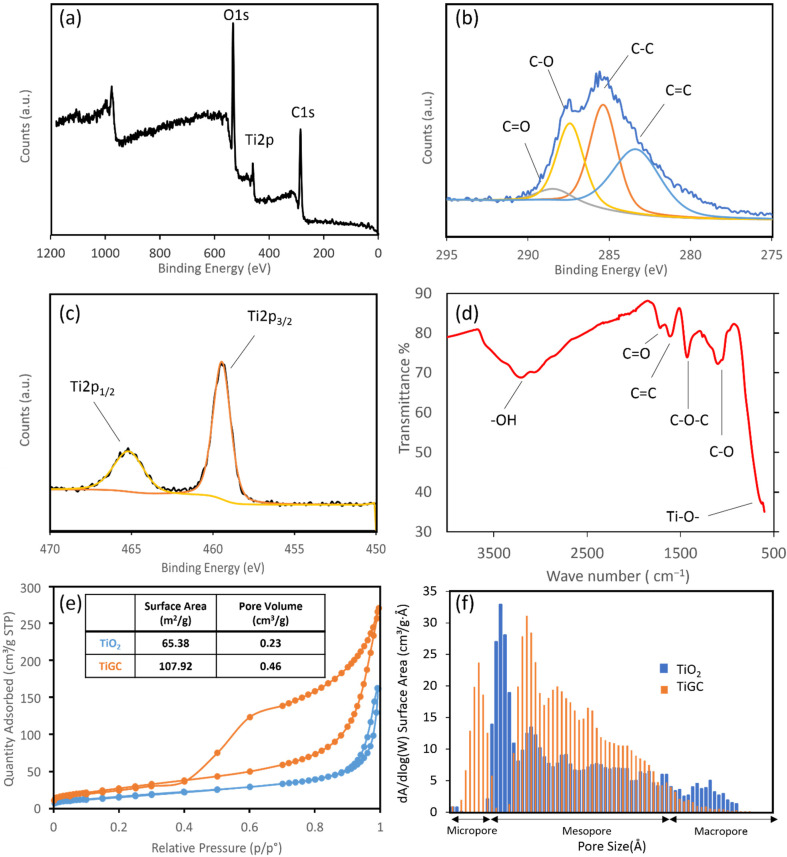
(**a**) Wide scan XPS spectrum of TiGC; (**b**) C 1s XPS spectrum of TiGC; (**c**) Ti 2p XPS spectrum of TiGC; (**d**) FTIR spectra of TiGC; (**e**) nitrogen adsorption–desorption isotherms, specific surface area (SSA), and (**f**) pore size distribution of TiGC and TiO_2_.

**Figure 3 nanomaterials-11-02087-f003:**
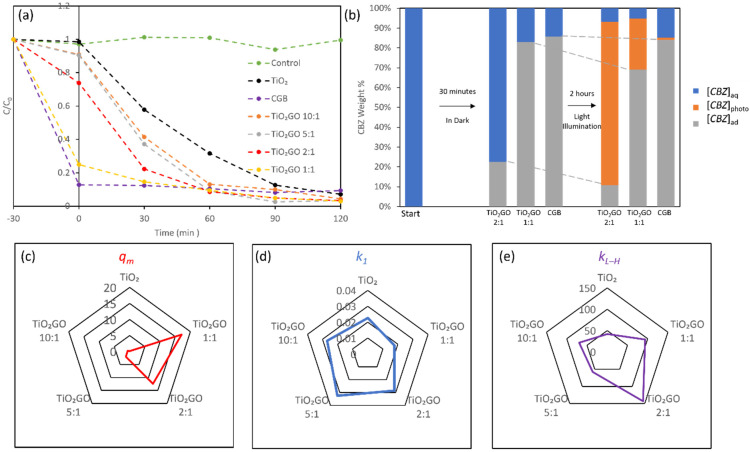
(**a**) C/C_0_ plot for CBZ photodegradation in the absence of catalyst (control) and using TiO_2_, CGB, and TiGC as catalysts (different TiO_2_:GO ratios 10:1, 5:1, 2:1, and 1:1. CBZ initial concentration = 1 mg/L, catalyst dose = 0.2 mg/mL; (**b**) the weight proportions of CBZ remaining in aqueous phase [*CBZ*]_aq_, adsorbed via TiGC [*CBZ*]_ad_, and photodegraded via TiGC [*CBZ*]_photo_ at different experiment times; (**c**) the comparison of Langmuir maximum adsorption capacity *q_m_* (mg/g); (**d**) the comparison of pseudo first-order reaction constant *k*_1_ (min^−1^); (**e**) the comparison of L-H model reaction constant *k_L-H_* (µg L^−1^ min^−1^).

**Figure 4 nanomaterials-11-02087-f004:**
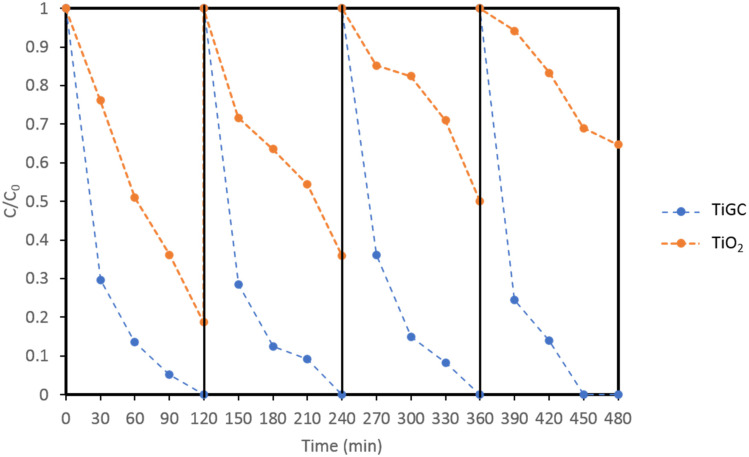
Comparison of CBZ removal by TiGC (TiO_2_:GO ratio 2:1) and bare TiO_2_ over repeated cycles of CBZ addition. CBZ initial concentration per cycle: 1 mg/L, 1000 W Xenon lamp supplied illumination.

**Figure 5 nanomaterials-11-02087-f005:**
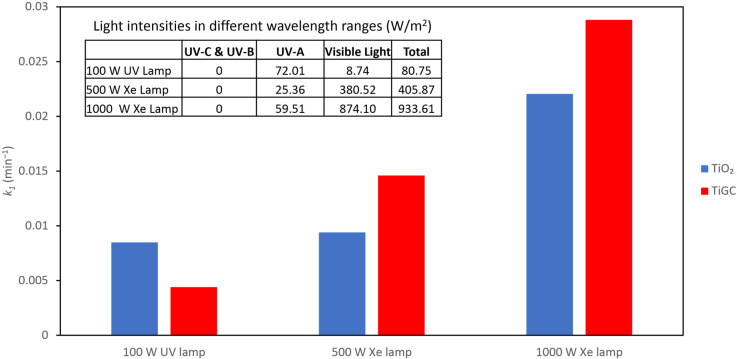
Pseudo first-order reaction constant *k*_1_ of CBZ photodegradation under different light intensities. CBZ initial concentration = 1 mg/L; catalyst dose TiO_2_ and TiGC (TiO_2_:GO ratio 2:1) = 0.2 mg/mL.

**Figure 6 nanomaterials-11-02087-f006:**
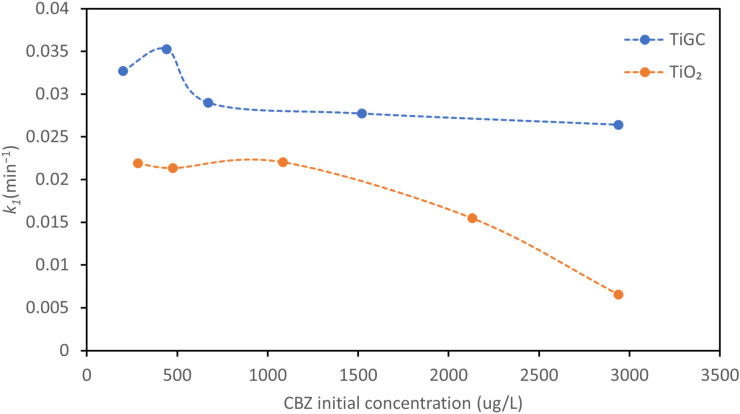
Pseudo first-order reaction constant *k*_1_ of CBZ photodegradation under different CBZ initial concentrations. Catalyst dose TiO_2_ and TiGC (TiO_2_:GO ratio 2:1) = 0.2 mg/mL. 1000 W Xe lamp supplied light.

**Figure 7 nanomaterials-11-02087-f007:**
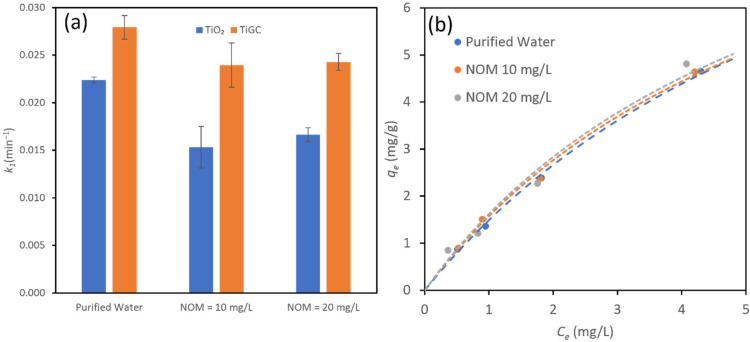
(**a**) Pseudo first-order reaction constant, *k*_1_, of CBZ photodegradation by TiO_2_ and TiGC (TiO_2_:GO ratio 2:1) under different NOM concentrations (Control for purified water, 10 mg/L NOM and 20 mg/L NOM). CBZ initial concentration = 1 mg/L, catalyst dose = 0.2 mg/mL. The error bar represents the standard deviations of triplicate experiments; (**b**) the adsorption isotherms of TiGC (TiO_2_:GO ratio 2:1) under different NOM concentrations. The dashed line represents the Langmuir model fit.

**Figure 8 nanomaterials-11-02087-f008:**
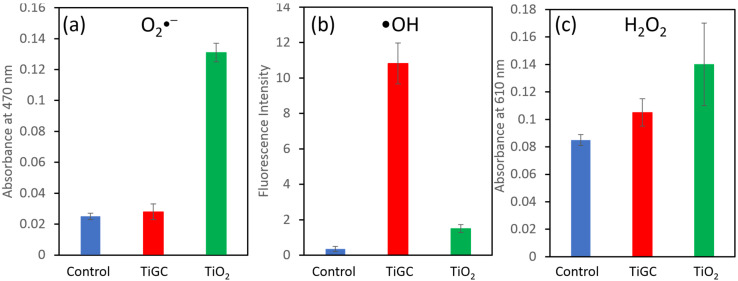
Production of (**a**) superoxide anion, (**b**) hydroxyl radical, and (**c**) hydrogen peroxide formed by 10 mg/L TiO_2_ and TiGC (TO_2_:GO ratio 2:1) exposed to 1000 W Xe lamp for 30 min. The error bar represents the standard deviation of triplicate experiments.

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
