# Peer review of "TiO2 (Core)/Crumpled Graphene Oxide (Shell) Nanocomposites Show Enhanced Photodegradation of Carbamazepine"

_nanomaterials, 2021, doi:10.3390/nano11082087_

Round 1

Reviewer 1 Report

The paper titled “ TiO2 (Core)/Crumpled Graphene Oxide (Shell) Nanocomposites Show Enhanced Photodegradation of Carbamazepine” reported the synthesis, characterization of TiO2/Graphene oxide for the photodegradation of Carbamazepine.

The paper is well written and the structure is correct, however major revision are requested before possible publication in the journal.

  1. What is the novelty of this work?, please cite it carefully in the abstract
  2. Only the interesting results, particularly for the adoptive properties should be supplemented in the abstract. Please add the capacity and adoption rate, repeatability obtained for the prepared materials.

Introduction:

  1. The introduction is low and not organized. Works on the Graphene oxide must be discussed in the introduction to show the choice of this material.
  2. Authors should discuss the properties, particularly the adsorptive properties of Gox and TiO2.

Results and discussion

  1. Line 188: Why the TiGC particle size increased when we increase the TiO2 amount? Please explain, scientifically this results.
  2. Line 211: The interpretation of specific surface area analysis is low and need to be rewritten: please explain the increases or decreases of surface area? It should be coherent with the pore size? It will be interesting to explain the phenomenon on the increases of surface area and pore size.
  3. What about the pore volume?
  4. The results on the CBZ photodegradation are interesting and well organized, but always some weaknesses on the scientific explanation is visible, if the authors car try to explain more the results with more references.
  5. In order to show the originality of this work, it will be interesting to compare the results obtained for TiGC with other sophisticates materials.

Reviewer 2 Report

  1. The authors demonstrated the photodegradation process of Carbamazepine by TiGC to obtain less harmful products. Could authors provide more information in this aspect? For example, what are the carbon-containing products after degradation? Or the schematic diagram of degradation mechanism. Various photodegradation products of carbamazepine may be identified by mass spectrometry or other techniques.
  2. In Graphical Abstract section(page2/16), it seems that the ordinate this figure couldn’t reflect the information, as should be done with reference to figure 7 (a).
  3. In page 9/16, the authors pointed out “The regeneration of TiGC (ratio 2:1) composite was also investigated by repeating the addition new CBZ solution for four cycles (Fig 4).” It seems that the “regeneration” isn’t suitable here, which means that the regeneration process of TiGC couldn’t be verified by this experiment.
  4. In Fig 4, the 4th cycle of TiGC seems to have better results than the previous three times, could authors explain the reasons.
  5. The stability of the catalyst is crucial. Could authors provide more information in this aspect? It seems that 4 cycles isn’t enough.
  6. After the photodegradation, is there any changes of the TiGC? Compared with other types of catalysts, the morphology of core-shell catalysts is more likely to change before and after reaction. Could authors provide more information.
  7. In photocatalytic reactions, pH is one of the most important parameters that influence the degradation of pollutants. Could author provide more information on the effect of pH?

Round 2

Reviewer 1 Report

Now the revised paper can be accepted in its present form.

Reviewer 2 Report

The paper can be accepted now